# Combined immunoinformatic approaches with computational biochemistry for development of subunit-based vaccine against *Lawsonia intracellularis*

Zahed Khatooni[1]*, Gordon Broderick[1], Sanjeev K. Anand[2], Heather L. Wilson[1,3,4]*

**1** Vaccine and Infectious Disease Organization (VIDO), University of Saskatchewan, Saskatoon, Saskatchewan, Canada, **2** Now with Modulant Biosciences LLC, Fishers, IN, United States of America, **3** Department of Veterinary Microbiology, Western College of Veterinary Medicine, University of Saskatchewan, Saskatoon, Saskatchewan, Canada, **4** School of Public Health, Vaccinology & Immunotherapeutics program, University of Saskatchewan, Saskatoon, Saskatchewan, Canada

* zahed.khatooni@usask.ca (ZK); heather.wilson@usask.ca (HLW)

**Data Availability Statement:** All relevant data are within the paper and its Supporting Information files.

## Abstract

*Lawsonia intracellularis* (LI) are obligate intracellular bacteria and the causative agent of proliferative hemorrhagic enteropathy that significantly impacts the health of piglets and the profitability of the swine industry. In this study, we used immunoinformatic and computational methodologies such as homology modelling, molecular docking, molecular dynamic (MD) simulation, and free energy calculations in a novel three stage approach to identify strong T and B cell epitopes in the LI proteome. From ∼ 1342 LI proteins, we narrowed our focus to 256 proteins that were either not well-identified (unknown role) or were expressed at a higher frequency in pathogenic strains relative to non-pathogenic strains. At stage 1, these proteins were analyzed for predicted virulence, antigenicity, solubility, and probability of residing within a membrane. At stage 2, we used NetMHCPan4-1 to identify over ten thousand cytotoxic T lymphocyte epitopes (CTLEs) and 286 CTLEs were ranked as having high predicted binding affinity for the SLA-1 and SLA-2 complexes. At stage 3, we used homology modeling to predict the structures of the top ranked CTLEs and we subjected each of them to molecular docking analysis with SLA-1*0401 and SLA-2*0402. The top ranked 25 SLA–CTLE complexes were selected to be an input for subsequent MD simulations to fully investigate the atomic-level dynamics of proteins under the natural thermal fluctuation of water and thus potentially provide deep insight into the CTLE-SLA interaction. We also performed free energy evaluation by Molecular Mechanics/Poisson−Boltzmann Surface Area to predict epitope interactions and binding affinities to the SLA-1 and SLA-2. We identified the top five CTLEs having the strongest binding energy to the indicated SLAs (-305.6 kJ/mol, -219.5 kJ/mol, -214.8 kJ/mol, -139.5 kJ/mol and -92.6 kJ/mol, respectively.) W also performed B-cell epitope prediction and the top-ranked 5 CTLEs and 3 B-cell epitopes were organized into a multi-epitope subunit antigen vaccine construct joined using EAAAK, AAY, KK, and GGGGG linkers with 40 residues of the LI DnaK protein attached to the N-terminus to further enhance the antigenicity of the vaccine construct. Blind docking studies showed strong interactions between our vaccine construct with swine Toll-like

**Funding:** This work has been supported by a Collaborative Research Grant (423278) to Dr. Heather L. Wilson. The funder had a role in study design and data analysis.

**Competing interests:** The authors have declared that no competing interests exist.

receptor 5. Collectively, these molecular modeling and immunoinformatic analyses present a useful *in silico* protocol for the discovery of candidate antigen in many viral and bacterial pathogens.

## Introduction

*Lawsonia intracellularis* (LI) are obligate intracellular bacteria that are the causative agents behind proliferative enteropathy (PE) [1–3] and which manifests as proliferative hemorrhagic enteropathy (PHE) and porcine intestinal adenomatosis (PIA) diseases in pigs [4–8]. Antibiotics are often used to reduce prevalence of LI infections but with the global reduction in use of sub-therapeutic antibiotics in livestock, alternatives to antibiotics are sought [9]. An effective subunit LI vaccine does not currently exist for this disease, and it is an area of study [10–13]. Developing a safe and effective subunit vaccine against LI can reduce suffering, decrease the economic impact of the disease on the swine industry and limit the usage of antibiotic use in pigs.

To generate a CD8+ T cell response to an vaccine antigen, antigen-presenting cells (APCs) must take up the vaccine and the antigens must escape the endosome to reside in the cytosol [14]. From here, they are directed to the cell's proteasome where they are processed into peptide fragments that are transported into the endoplasmic reticulum (ER) [15, 16]. Here, the peptide fragments are loaded on the peptide binding groove of major histocompatibility complex I (MHCI)/swine leucocyte antigen complexes (SLA-1/2) [16, 17]. The APCs migrate to draining lymph nodes to present the cytotoxic T cell epitopes (CTLEs) to T cell receptors (TCRs) on CD8+ T cells. Upon cognate TCR-CTLE/SLA recognition, a T cell response is initiated which may include lymphocyte proliferation and increased antigen-specific cytokine production [18, 19]. How strongly a CTLE interacts with the SLA peptide binding groove is a critical aspect in determining which T-cell epitopes are the most effective at inducing a T cell response. Predicting CTLEs that have high binding affinity to SLA peptide binding grooves remains extremely challenging.

Our focus was to use computational approaches as a rational strategy to determine which CTLEs in the LI proteome have high affinity for the SLA peptide binding groove site and therefore are likely most effective to induce a robust cell mediated immune response when they are part of a subunit vaccine. Our approach consists of three stages for epitope selection wherein, in the initial two stages, we use a series of commonly known artificial intelligence and machine learning tools to perform immunoinformatic assessments. We selected representative SLA-1 and SLA-2 alleles that have known crystal structures. We then parsed the LI proteome to identify proteins that are highly expressed in pathogenic LI and have unknown function, which would become our antigen pool. The antigens were subjected to immunoinformatic analyses to predict and rank their antigenicity, virulence, solubility and to map their secondary structures. In the CTLE prediction stage, we used NETMHCpan4.1 to generate several thousand epitopes and rank them based on their predicted binding affinity to the two SLA alleles [20–22]. The top ranked CTLEs were subjected to further online tools to predict and rank them based on antigenicity, allergenicity, toxicity and susceptibility to common peptidases, as a measure of epitope stability. The novelty of our approach lies in the validation of these tools in the post-evaluation stage. We modeled the conformation of the top ranked CTLEs then performed docking studies with the 2 SLA complexes to calculate the free binding energy as a mean to quantify binding affinity. We then utilized molecular dynamics (MD) simulations to fully investigate the atomic-level dynamics of proteins under the natural thermal fluctuation of

water and thus potentially provide deep insight into the CTLE-SLA interaction (Fig 1). Our work is among very few work that combine extended MD simulations and free energy evaluation with database informed immunoinformatic to enhance *in silico* epitope prediction. We performed extensive bioinformatic analysis to predict B cell epitopes then we constructed a multi-epitope protein construct consisting of 5 CTLEs and 4 B cell epitopes that can be assessed as a vaccine subunit antigen. We expect that future animal trials will confirm that this multi-epitope synthetic antigen will elicit a strong cytotoxic T cell and antibody-mediated immune response, and that this *in silico* approach to antigen design offers a highly efficient means of reducing the vaccine development cycle time.

## Material and methods

### Stage 1-SLA and antigen selection

**SLA protein analysis.**   Out of the 100 SLA-1 and 105 SLA-2 complexes, we selected SLA-1*0401 (PDB: 3QQ3) [23] and SLA-2*040202 (PDB: 6A6H) [24] as the representative SLA alleles [25] because they are highly expressed in many pig populations and their structures are in Uniprot [26, 27].

**Antigen selection pool.**   Using Uniprot [26, 27] and PubMed, we identified $\sim$ 1342 proteins from the reference LI proteome from PHE/MN1-00 strain (uniprot.org/proteomes/UP000002430). From these, we included only 256 proteins with the criteria that they have

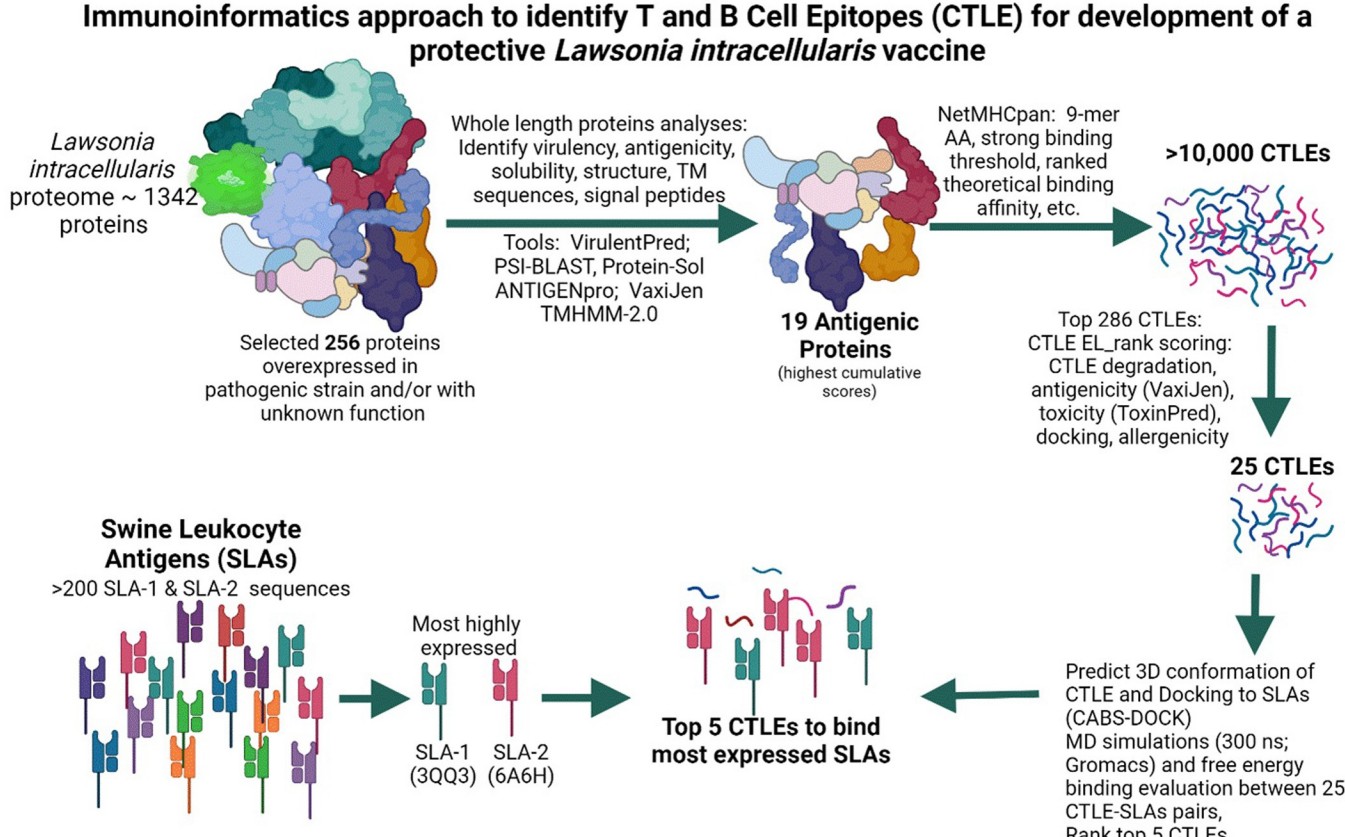

**Fig 1. Schematic of computational prediction approach used to identify the optimal T cell epitopes bind to highly expressed pig SLA molecules.** Created in BioRender. Wilson, H. (2022) BioRender.com/x56b967.

unknown function and they are predicted to be full length proteins and/or they were over-represented in pathogenic LI strains, based on research by Vanucci et al. [25].

**Antigen selection.** The primary amino acid sequences of the 256 LI proteins of interest were submitted to the VirulentPred platform using default parameters to predict virulent proteins within the bacterial protein repertoire (http://bioinfo.icgeb.res.in/virulent/) using residue composition, dipeptide composition, and higher order dipeptide composition. Next, they were subjected to PSI-BLAST using default parameters to generate position-specific scoring matrix (PSSM) The 256 LI proteins were next assessed for solubility using Protein-Sol (http://protein-sol.manchester.ac.uk) with antigens assigned a higher than 0.45 predicted solubility ranked as being soluble. The protein cohort was then submitted to VaxiJen Platform (http://www.jenner.ac.uk/VaxiJen) (Antigenicity I) and ANTIGENpro [28, 29] (http://scratch.proteomics.ics.uci.edu) (Antigenicity II). The protein cohort was processed with the TMHMM-2.0 algorithm to identify proteins predicted to have transmembrane (TM) helices. Of the 256 candidate proteins, 19 were selected as most promising for continued analysis, according to the cumulative scores and assessment criteria established for CTLE prediction.

## Stage 2-Epitope prediction

**CTLE prediction based on analysis from full-length proteins.** From 19 selected full-length antigens, there are several thousand potential CTLEs present with a length of 8–12 amino acids [20]. To predict the optimal epitopes, the 19 selected proteins were submitted to NetMHCpan-4.1 [20–22] using the following parameters: peptide length: 9 aa, threshold for strong binder: 0.5 (% rank), threshold for weak binder: 2 (% rank), inclusion of theoretical binding affinity (predicted $IC_{50}$ values). All selected CTLE were ranked for antigenicity determined again using the VaxiJen platform with a predicted relative antigenicity threshold of 0.40 [30]. CTLEs were then submitted to the Protein Digest server (http://db.systemsbiology.net:8080/proteomicsToolkit/proteinDigest.html) to determine which CTLEs were sensitive to digestion by common proteases such as trypsin and chymotrypsin. CTLEs were finalized if they have no sign of toxicities for the host organism as reported by ToxinPred [31] using default parameter values. We subjected the top CTLEs to AllerCatPro.2 (doi.org/10.1093/nar/gkac446) and determined that none of them showed any significant hit for allergenicities [32].

## Stage 3 -Post evaluation of CTLE peptides

**Structure prediction by homology modelling, structural assessment, binding site selection and molecular docking.** The 3D conformations of the CTLE were generated by CABS-Dock [33–35]. Ramachandran plots were generated using PROCHECK (https://www.ebi.ac.uk/thornton-srv/software/PROCHECK/) using default parameters [36].

These CTLEs were subjected to molecular docking analysis with SLA-1*0401 [23] and SLA-2*0402 [24] also using CABS-Dock (https://bitbucket.org/lcbio/cabsdock) to find the best poses between each SLA and CTLE. The top ranked 25 SLA–CTLE complexes with the closest pose to the crystallized complexes of SLA-1 and SLA-2 with previously showing higher EL_rank scores were selected to be an input for subsequent MD simulations [37].

**Molecular dynamic simulation.** The MD simulation of CTLE-SLA complexes for all of the 25 CTLEs that docked with SLA-1 and SLA-2 was performed for 300 ns using Gromacs molecular dynamic package (Gromacs 5, 2020 and 2023), and all atoms CHARMM36 force field [38, 39]. For each set of simulations, a cubic box with periodic boundary conditions and 1.0 nm distances between the box edge and SLA–CTLE complexes were defined before it was fully filled out by TIP3P water models in every 25 boxes. Systems were neutralized and all box molecules were energy minimized by the steepest descent minimization algorithm. After

minimization, all conformations were compared with the initial coordinates to ensure there are relaxed and optimized complexes with no atomic clashes or unfavorable contacts. The time step for each simulation was set to 2 fs, and "md" integrator was employed to integrate Newton's equations of motion. The electrostatic (long-range) and van der Waals (short-range) interactions were treated by Particle Mesh Ewald and Lennard Jones, respectively, while applying a 1.2 nm cut off. The temperature and pressure were kept stable at 300 k and 1 bar by assigning modified Berendsen thermostat [40, 41] (V-rescale) with time constant τ_t = 0.1 ps and Parrinello-Rahman pressure coupling with the compressibility of 4.5e-5 and τ_ p = 2 ps. The LINCS algorithm was used as the constraint algorithm for bond length in all simulations. Before the production run for 300 ns the Position Restrain (PR, NVT ensemble (constant Number of particles, Volume, and Temperature)) and NPT simulation (NPT ensemble (constant for Number of particles, Pressure, and Temperature)) were conducted for 1 ns and 5 ns, respectively, with the same 2 fs time step. The 5 ns of NPT simulation was performed after the NVT simulation. MD calculations (NVT, NPT and production simulation) were submitted to Compute Canada (https://www.computecanada.ca/) and were performed on the Graham supercomputing cluster (The Digital Research Alliance of Canada). Tools and software for visualization, or generating graphs included VMD (https://www.ks.uiuc.edu/Research/vmd), PyMol (https://pymol.org/2/) and Grace (https://plasma-gate.weizmann.ac.il/Grace/) were used.

## Free energy evaluation

Free energy of binding between SLAs and each CTLE was calculated by using *g_mmpbsa* tool from 250 ns – 300 ns with -dt set to 100 ps. This method was developed based on molecular mechanics Poisson–Boltzmann surface area (MM-PBSA) [42, 43].

The binding free energy is measured as

$$\Delta G_{binding} = G_{SLA-CTLE} - (G_{SLA} + G_{CTLE})$$

Eq : 1

The $G_{SLA - CTLE}$ is the total energy between SLA and CTLEs. The free energy in general was evaluated according to Eq 2[42].

$$G_z = \langle E_{MM} \rangle - TS + \langle G_{Solvation} \rangle$$

Eq : 2

The z is an indication for protein-protein and $\langle E_{MM} \rangle$ counts average vacuum potential energy, which is the potential energy for both bonded and nonbonded interactions[42]. The nonbonded interaction includes electrostatic and van der Waals were calculated using the *g_mmpbsa* [42] tool. The bonded interaction measures the contribution of angles, bonds, dihedrals and improper interactions. The entropy (S) and temperature (T) are evaluated as the TS, which defines the entropic contribution in Eq 2, and $\langle G_{Solvation} \rangle$ is the average free energy of solvation as it requires to move solute from a vacuum into the solvent.

## B-cell epitope predictions

Four proteins were selected for the prediction of B-cell epitopes, two with high expression in the pathogenic strain including outer membrane efflux protein (Uniprot ID: Q1MPM8) and Hydrogenase maturation factor (Uniprot ID: Q1MRS4), as well as two other proteins consisting of the LI autotransporter protein A (Uniprot ID: Q1MQM4) that has been shown as a potential antigen and the Chaperone protein DnaK (Uniprot ID: Q1MPW1) [25, 44–47]. To identify the optimal linear and conformational B cell epitopes from these antigens, these four proteins were analyzed using BepiPred- 2.0 (with a selected threshold of 0.5), Bcepred (https://webs.iiitd.edu.in/raghava/bcepred/index.html) and likewise with ABCpred. The 3D

conformations of the four proteins were modelled by Robetta [48, 49], and Protein Homology/ analogY Recognition Engine V 2.0 (Phyre2) were used to predict the continuous conformational B-cell epitopes. Results from the different platforms were compared to identify the regions predicted to be strong B cell epitopes which were then selected to include in the subunit antigen.

## Modelling of vaccine construct and CTLE interaction with TLR-5

Robetta was used to predict the 3D conformation of the whole subunit antigen [50] and the 3D conformation of TLR-5 was predicted by means of SWISS-MODEL (http://swissmodel. expasy.org/) [51]. We assessed the interaction between the vaccine construct and the pig TLR5 (Uniprot: T1UMR1_PIG) using HDOCK with default criteria.

# Results and discussion

## Selection of representative SLA-1 and SLA-2 alleles

SLA class I proteins are comprised of SLA-1, SLA-2, SLA-3 and SLA-6 alleles. We selected SLA-1*0401 [23] and SLA-2*040202 [24] to represent the SLA class I alleles because their crystal structures are available (PDB 3QQ3 and 6A6H, respectively) and they are highly expressed in the pig population. In **Fig 2**, we compare the primary sequences for SLA-1 and SLA-2 using JalView. The amino acids (AAs) that comprise the α1, α2 and α3 domains are AA1-90, AA91-180 and AA 180–275, as indicated in **Fig 2** and they show 87% sequence similarity [52]. Despite this high degree of similarity, our homology modeling of SLAs has previously shown that while the sequences and conformation of the overall SLA and/or the peptide binding groove may appear nearly identical, the changes to the electrostatic contact mapping may be dramatically different with just a few amino acid differences [20]. Therefore, homology modeling followed by molecular docking and MD simulation are critical to predict CTLE-SLA peptide binding groove affinities.

## Pre-prediction stage to identify possible antigens from the *L. intracellularis* proteome

The reference proteome for the PHE/MN1-00 strain (uniprot.org/proteomes/UP000002430) is predicted to have ∼ 1342 proteins using Uniprot Proteome [26, 27] and PubMed (data not shown). We manually parsed the proteins to select those that are highly expressed in pathogenic LI strains based on research by Vanucci et al. [25] or were proteins with unknown function (256 proteins listed in **S1 Table in S1 File**). These proteins with unknown function may be critically required for growth, adaptation to nutritional status, and virulence, making them suitable vaccine antigen candidates [53, 54]. We selected proteins with unknown function to avoid any patent infringement if vaccine development will be pursued and because we wanted to show that our approach of epitope selection does not require proteins with known structure or function.

The 256 selected LI proteins were then submitted to the VirulentPred platform to predict bacterial virulent proteins within the protein repertoire. Output from this tool includes residue composition (Resn), dipeptide composition, and higher order dipeptide composition, PSI-BLAST-PSSM, SVMs and PSI-BLAST columns. If values were >0.9 for any of these scores, they were marked with bold font indicating they met virulence for each criterion. They were ranked and then they were labeled as predicted to be virulent or non-virulent.

The 256 selected LI proteins were then subjected to immunoinformatic assessments using a series of artificial intelligence and machine learning tools to determine expected virulence,

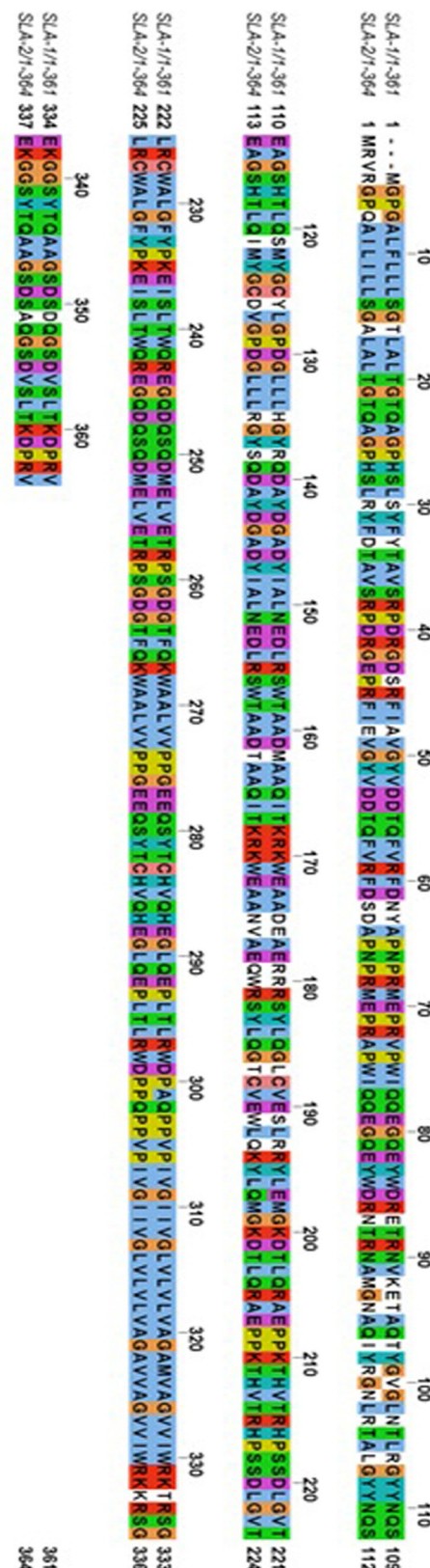

**Fig 2. Sequences variations between representative SLA-1 and SLA-2 alleles.** The sequences alignment between SLA-1 3QQ3 and SLA-2 6A6H share ~ 87% sequence similarity. The regions consisting of amino acids 1–90, 91–180 and 181–275 are referred to as α1, α2 and α3, respectively. The α1 and α2 domains comprise the peptide binding groove.

solubility, antigenicity, and potential transmembrane regions (Table 1). The solubility for 256 proteins were calculated using Protein-Sol which uses features largely based on bimodal distribution of entity solubilities for proteins in *Escherichia coli* in the context of cell-free expression. Proteins assigned > 0.45 predicted solubility were ranked as being soluble in solution. The sequences for all 256 proteins were then uploaded to ANTIGENpro (Antigenicity I threshold >0.4) [30] then VaxiJen (Antigenicity II, threshold >50%) [27, 29] to predict antigenicity based on the default parameters from these online tools. VaxiJen platform is an alignment-independent server that predicts which antigens in a cohort will generate a protective response solely based on the physicochemical properties (http://www.jenner.ac.uk/VaxiJen) (Antigenicity II). VaxiJen transforms the sequences after calculation of auto cross-covariance into a vector of principal amino acid properties with a minimum predictive accuracy of 70% based on the documented physicochemical properties of proteins [30]. ANTIGENpro [28, 29] uses core principle of protein sequence without any alignment and instead uses machine learning algorithms to estimate the probability of antigenicity. Proteins that generated > 0.4 relative antigenicity using VaxiJen and >50% using ANTIGENpro were selected as favorable. Finally, the

**Table 1.** *Lawsonia intracellularis* proteins selected based on pre-prediction criteria.

| UNIPROT Entry | Resn (>0.9) | Higher order (>0.9) | PSI-BLAST-PSSM | SVMs and PSI-BLAST (0.9) | Sol (0.45) | Antigenicity I (0.4) | Antigenicity II (%) | TMHMM |
|---|---|---|---|---|---|---|---|---|
| Q1MP04 | -0.24 | 0.53 | **Virulent** | -0.73 | **0.74** | 0.31 | **0.85** | **0** |
| Q1MQK7 | **0.95** | **1.78** | **Virulent** | **1.05** | **0.75** | **0.89** | **0.86** | **0** |
| Q1MP64 | **1.24** | 0.43 | **Virulent** | **1.01** | **0.69** | **0.76** | **0.95** | ~9 |
| Q1MS98 | 0.84 | **1.02** | **Virulent** | **1.07** | **0.75** | **1.0** | **0.91** | **0** |
| Q1MPX4 | **1.34** | 0.66 | **Virulent** | **1.06** | **0.73** | **0.75** | **0.93** | **0** |
| Q1MP78 | 0.67 | 0.74 | **Virulent** | **1.04** | **0.73** | **0.85** | **0.93** | **0** |
| Q1MRZ2 | **0.92** | **1.02** | **Virulent** | **1.03** | **0.71** | **0.6** | **0.94** | **0** |
| Q1MNT3 | 0.33 | **1.16** | **Virulent** | 0.86 | **0.58** | **0.82** | **0.94** | **0** |
| Q1MPE6 | **1.01** | **1.77** | **Virulent** | **1.04** | **0.71** | **0.87** | **0.90** | **0** |
| Q1MP58 | 0.49 | 0.31 | **Virulent** | **1.07** | **0.89** | **0.88** | **0.93** | **0** |
| Q1MRS2 | **1.46** | **1.67** | **Virulent** | **1.10** | **0.80** | **0.72** | **0.77** | **0** |
| Q1MQK8 | **1.35** | **1.21** | **Virulent** | **1.10** | **0.64** | **0.74** | **0.87** | **0** |
| Q1MNQ8 | **1.39** | **2.11** | **Virulent** | **1.01** | **0.70** | **0.52** | **0.90** | **0** |
| Q1MSA8 | **0.90** | **1.16** | **Virulent** | **1.11** | **0.74** | **0.6** | **0.75** | **0** |
| Q1MS15 | -0.62 | -0.18 | Non-Virulent | -1.04 | **0.67** | **0.47** | 0.06 | **0** |
| Q1MRS4 | **1.05** | 0.12 | Non-Virulent | -0.37 | **0.66** | **0.41** | **0.55** | **0** |
| Q1MP82 | **0.93** | 0.55 | **Virulent** | **1.04** | **0.60** | 0.17 | 0.17 | **0** |
| Q1MR65 | -0.25 | 0.16 | Non-Virulent | -1.01 | **0.58** | **0.42** | **0.76** | **0** |
| Q1MRR1 | 0.79 | 0.78 | **Virulent** | **1.00** | 0.49 | **0.72** | 0.48 | **0** |

From a pool of 256 proteins LI proteins, 19 proteins were ranked as the highest based on assessment of virulency, solubility, antigenicity, and membrane helicity. Bolded font indicate that these proteins met the minimum threshold. Abbreviation used in the table. Resn: Residue composition. Higher order: Higher Order Dipeptide Composition Based. PSI-BLAST-PSSM: PSI-BLAST created PSSM Profiles. SVMs and PSI-BLAST: Cascade of SVMs and PSI-BLAST (all based on VirulentPred). Sol: solubility prediction-based Protein-Sol. Antigenicity I: VaxiJen based antigenicity prediction. Antigenicity II: ANTIGENpro based antigenicity detection. TMHMM: Transmembrane helices prediction based on TMHMM - 2.0.

protein cohort was processed with the TMHMM-2.0 algorithm to determine the secondary structures of the antigens [55]. Based on the physicochemical properties of proteins, Q1MP64 protein had epitopes with predicted transmembrane (TM) regions so this antigen was excluded as suitable vaccine candidates (**Table 1**).

## Epitope prediction phase

Among the 256 entries, the 19 proteins with the highest scores for virulence, solubility, allergenicity antigenicity, virulence, and did not have a transmembrane domain were selected for further evaluations. CTLEs Proteins including Q1MQK7, Q1MP64, Q1MRZ2, Q1MPE6, and Q1MNQ8, showed high scores in most or all assessments (**Table 1**). S1 Table in S1 File lists results from all 256 proteins. These results clarify how several factors can contribute to the predicted antigenicity of proteins within a bacterial proteome. At this time, there is no clear formula for predicting which antigens would contribute to an effective vaccine, but we used these online tools to give us best-guesses for promising bacterial antigens who's predicted epitopes can be interrogated and characterized below.

From 19 selected full-length antigens, there are several thousand potential CTLEs present with a length of 8–12 amino acids [20]. To predict the optimal epitopes for use in our vaccine, these 19 selected proteins were submitted to NetMHCpan-4.1 [20–22], which uses a NNAlign_MA machine learning framework to select for 9-mer peptides with predicted binding affinity to SLA-1 and SLA-2 [56, 57]. The output is EL_Rank Score and the lowest score indicates the most favorable binding of the CTLE to the SLA complexes. A total of 286 epitopes were identified that were predicted to have strong binding affinity for two representative SLA complexes (**Table 2**). Results showed that Q1MNQ8_a2 has one of the best EL-Rank with 0.0214 and CAJ54302_a7 had the lowest CTLE EL-Rank with 1.8788.

Next, the top 286 CTLEs were assessed for antigenicity using VaxiJen default parameters and those with >0.4 score were considered optimal (**Table 2**). Protein Digest service was used to predict digestion of the predicted CTLE based on presence of optimal cleavage sites for enzymes such as trypsin and chymotrypsin. Most of the CTLEs were very sensitive to enzyme cleavage however we opted to include the highly ranked CTLEs if digestion yielded 7 or 8 residues, because the optimal SLA binding length is 8–12 aa. If the epitope was predicted to be cleaved in such a way that the CTLE was highly disrupted, it was excluded from further analysis regardless of whether they ranked as predicted to be highly immunogenic. Next, the probability of being a signal peptide (S/P) was determined by submitting the CTLEs to SignalP-5.0 and only those without signal peptides were considered further. None of the epitopes showed allergenicity as assessed by AllerCatPro.2 (data not shown) [32]. Finally, the sequences were submitted to ToxinPred to predict whether highly expressed epitopes were toxic for organisms. None of the CTLEs were predicted to be toxic. The entire data set is shown in S2 Table in S1 File. These online resources describe predicted characteristics of the epitopes and tools such as NetMHCPan4.1 has its own mathematical formulas to predict binding affinity to only a limited number of SLAs and it can inform epitope selection. But again, there is no clear formula for predicting which epitopes will bind with strong affinity to MHCs, which is a critical component of predicting the strength of the resultant immune response. To predict binding affinity between the CTLE and the peptide binding groove, we needed to perform molecular docking and MD simulations.

## Post evaluation of CTLE peptides

The half-life of SLA-CTLE complex is the primary parameter that dictates the strength of the elicited T cell response [58]. Before we can predict the binding affinity of the peptide to the

**Table 2. Predicted CTLEs from 19 LI antigens ranked based on EL-rank, stability against common peptidases, presence of signal peptide and toxicity.**

| Rank | UNIPROT Entry | CTLE | EL_Rank | Antigenicity | Digestion | S-P | Docking | Charge | Mol Wt | Toxicity (T or NT) |
|------|---------------|------|---------|--------------|-----------|-----|---------|--------|--------|--------------------|
| 1 | Q1MNQ8_a2 | QLAPTPLLY | 0.0214 | 1.1814 | Y | N | Y | 0 | 1015.4 | NT |
| 2 | Q1MNQ8_a3 | ALEQQIHLM | 0.0704 | 0.6691 | N | N | Y | -0.5 | 1082.4 | NT |
| 3 | Q1MNQ8_a4 | QTQNTNTLF | 0.1171 | 0.4484 | N | N | Y | 0 | 1066.3 | NT |
| 4 | Q1MNQ8_a5 | TTNSQHPLF | 0.149 | 0.1511 | N | N | Y | 0.5 | 1044.3 | NT |
| 5 | Q1MNQ8_a6 | ASFINTETY | 0.2192 | -0.2724 | Y | N | | -1 | 1045.2 | NT |
| 6 | Q1MNQ8_a7 | STSMDGSGY | 0.2285 | 1.872 | Y | N | Y | -1 | 904.03 | NT |
| ... | ... | ... | ... | ... | ... | ... | ... | ... | ... | ... |
| 279 | Q1MP64_a1 | YVQSTAAMF | 0.1966 | -0.0353 | Y | N | Y | 0 | 1017.3 | NT |
| 280 | CAJ54302_a1 | NVETGREFY | 0.1418 | 0.7103 | Y | N | | -1 | 1114.3 | NT |
| 281 | CAJ54302_a2 | NIVPDSLQF | 0.2173 | 0.4796 | Y | N | | -1 | 1032.3 | NT |
| 282 | CAJ54302_a3 | KLQRVKVCY | 0.5144 | 0.3031 | Y | N | Y | 3 | 1136.6 | NT |
| 283 | CAJ54302_a4 | SLEGAILEI | 0.7529 | 1.1080 | N | N | | -2 | 944.23 | NT |
| 284 | CAJ54302_a5 | KIFTEGTSL | 0.767 | -0.2938 | Y | N | Y | 0 | 995.27 | NT |
| 285 | CAJ54302_a6 | VPDSLQFAF | 1.0915 | 0.7737 | Y | N | | -1 | 1023.3 | NT |
| 286 | CAJ54302_a7 | EIEKIPLML | 1.8788 | 0.6124 | Y | N | Y | -1 | 1085.5 | NT |

CTLEs were estimated by NetMHCpan-4.1 and organized by EL_Rank. Low EL_Rank scores indicate high favourability. The top 286 CTLEs were assessed for antigenicity, digestion sensitivities signal peptide estimation, docking, charge, molecular weight, and toxicities. The complete data from all high ranked epitopes are shown in S2 Table in S1 File. The antigenicity was evaluated by VaxiJen with the 0.4 threshold. Digestion by most common enzymes such as trypsin and chymotrypsin were marked with Y but if the cleaved CTLE was still at least 7 AA in length, it was deemed favourable. S-P column indicate whether the epitope is part of the signal peptide or not (N). Docking analyses was only performed for epitopes that were not highly cleaved by digestion enzymes- as indicated by Y. ToxinPred was used to predict whether highly expressed epitopes were toxic for organisms.

peptide binding grooves, we must accurately predict the 3-D conformations of the CTLEs then perform molecular docking with SLA complexes. We generated the 3D conformations of the CTLE using CABS-Dock [33–35]. In (**S1 Fig**), we show the conformations of crystalized epitopes in the Protein DataBank. In **S2 and S3 Figs**, we show the conformations of representative CTLEs and the corresponding Ramachandran plots.

The CTLEs were subjected to molecular docking with SLA-1*0401 [23] and SLA-2*040202 [24], also using CABS-Dock. The top 10 higher ranked docking poses for each CTLE were evaluated as they represent the best predicted energy binding to our selected SLAs. The top ranked 18 SLA-1-CTLE complexes (C1-C18) and the top 7 SLA-2-CTLE complexes (D1-D7) with the closest pose to the crystallized complexes SLA-1*0401 (PDB: 3QQ3) and SLA-2*040202 (PDB: 6A6H), respectively are shown in **Table 3** [23, 24]. **S4 Table in S1 File** shows the top three clusters for each selected CTLEs, their Root Mean Square Deviation (RMSD) and number of docked models in each of the top three clusters. Thus, we used molecular docking to computationally achieve an optimized conformation and relative orientation between the CTLE and the SLA peptide binding grooves [59]. The top ranked CTLE-SLA partners determined to have the lowest free energy were used as input for subsequent MD simulations [37].

Our next steps required that we track the atomic interaction between the amino acids that comprise the CTLEs and the peptide binding grooves over time. MD simulations refine initial docking models, ensuring that the epitopes adopt more biologically accurate conformations while revealing which interactions remain stable across the simulation period. We applied a combination of MD simulations and free energy analyses to investigate the molecular mechanism of SLA-peptide conformational diversity and to assess the strength of interactions between each epitope and the selected SLA. To the best of our knowledge, this study is among the few works that applied large-scale MD simulations and free energy evaluation on SLA–

Table 3. Top 25 ranked epitopes based on molecular docking simulations.

| Name | UNIPROT Entry | Selected CTLE | SLAI | | Name | UNIPROT Entry | Selected CTLE | SLAI |
|------|---------------|---------------|------|---|------|---------------|---------------|------|
| C1 | Q1MNQ8 | ALEQQIHLM | SLA-1 | | C14 | Q1MS15 | HYDASGLRF | SLA-1 |
| C2 | Q1MP64 | YVQSTAAMF | SLA-1 | | C15 | Q1MRR1 | ICIQCGHPF | SLA-1 |
| C3 | Q1MS98 | SSGSSGSHF | SLA-1 | | C16 | Q1MRS4 | SLEGAILEI | SLA-1 |
| C4 | Q1MP78 | TTSSHHGPY | SLA-1 | | C17 | Q1MPE6 | SVNESLIGF | SLA-1 |
| C5 | Q1MRZ2 | SIDTIPLQF | SLA-1 | | C18 | Q1MS15 | QLNVIEGHY | SLA-1 |
| C6 | Q1MNT3 | KIDVSPNEF | SLA-1 | | | | | |
| C7 | Q1MNT3 | TSEAGSHSL | SLA-1 | | D1 | Q1MQK8 | ENSNSGYSY | SLA-2 |
| C8 | Q1MPE6 | QVASQANQM | SLA-1 | | D2 | Q1MNT3 | HQVNVHFQY | SLA-2 |
| C9 | Q1MP58 | KQITPMMLL | SLA-1 | | D3 | Q1MR65 | ITMEELKEY | SLA-2 |
| C10 | Q1MSA8 | QVDTSTNNI | SLA-1 | | D4 | Q1MPX4 | ISFSTIISY | SLA-2 |
| C11 | Q1MQK8 | MVTSIQSSL | SLA-1 | | D5 | Q1MSA8 | KNINLQGGY | SLA-2 |
| C12 | Q1MQK7 | TVVHPSTPL | SLA-1 | | D6 | Q1MP58 | VRITLAQGF | SLA-2 |
| C13 | Q1MP04 | LSNTTSVKL | SLA-1 | | D7 | Q1MNQ8 | QLAPTPLLY | SLA-2 |

Top 18 highly-ranked CTLEs that bind SLA-1 (PDB: 3QQ3) and top 7 highly ranked CTLEs for SLA- 2 (PDB: 6A6h) are named from C1 –C18, and D1 –D7, respectively.

CTLE complexes to identify strong CTLEs. We performed 300 ns analyses on the top 25 ranked epitopes predicted to interact with the binding-site of the indicated SLA. Root Mean Square deviations (RMSD) (Fig 3) and Solvent Accessible Surface Area (SASA) (Fig 4) were used to assess peptide-SLA stability, and free binding energies as a measure of binding affinity between CTLEs and SLAs. For all analyses, the last 50 ns of MD simulations are shown, and for free energy, the - dt of 100 ps from 250 ns to 300 ns were selected.

The peptide binding site is comprised of the α1 (1–90 amino acids) and α2 (91–180 amino acids) regions of the SLA-1 and SLA-2 proteins and hence the constituent residues play a critical role in interacting with the CTLE. The stability of SLA-1 and SLA-2 α1 and α2 domains binding with the top 25 predicted CTLE were monitored by measuring the RMSDs in each simulation (Fig 3A–3D). In general, the average RMSD for simulations C2 - C6, C10, C12, C13, C16, C17, C18, are relatively stable and show convergence at around ∼ 3–6 Å. The RMSD value for C1, C8, C14, D1, D2 and D5 show large fluctuation over the route of the MD ∼ 8–9 Å; and these complexes might not be optimal choice for more processing, therefore, apart from D5 these will be exclude from the final selection. As compared to SLA-1 (C1 – C18), the SLA-2 simulations demonstrated larger RMSD for around 2–3 Å (Fig 3D).

Next, the stability of the SLA-CTLE for each simulation was assessed using SASA, which indicates the extent to which the surface area of molecules are accessible to interact with solvent and it provides information about the impact that each CTLE has on each SLA conformational integrity during the course of MD simulations. The SASA was evaluated using the reference structure set to the last atomic coordinates of the NPT simulations. SASA areas for C3 - C6 show SLA-1 SASA values from a min of ∼ 168 nm$^2$ to 190 nm$^2$, with fluctuation of ∼ 12 nm$^2$, whereas SAS areas for C1, C2, C7 and C11 showed relatively higher values with the average SASA values being higher than 185 nm$^2$ (Fig 4A and 4B). The SAS areas for C9, C10 and C12 are at the range of 175 nm$^2$- max of 195 nm$^2$. SAS area for C13 –C18 and D1 -D7 show average of lower than 198 nm$^2$ with min of ∼ 173 nm$^2$ in C14 and ∼ 177 nm$^2$ in D6 and larger than of 180 nm$^2$ to max of 198 nm$^2$ in C13, C15 –C18, D1 –D5 and D7 (Fig 4C and 4D).

RMSD and SASA evaluate the approximate stability and conformational integrity between the CTLE and the SLA peptide binding groove, whereas the MD simulation results showed

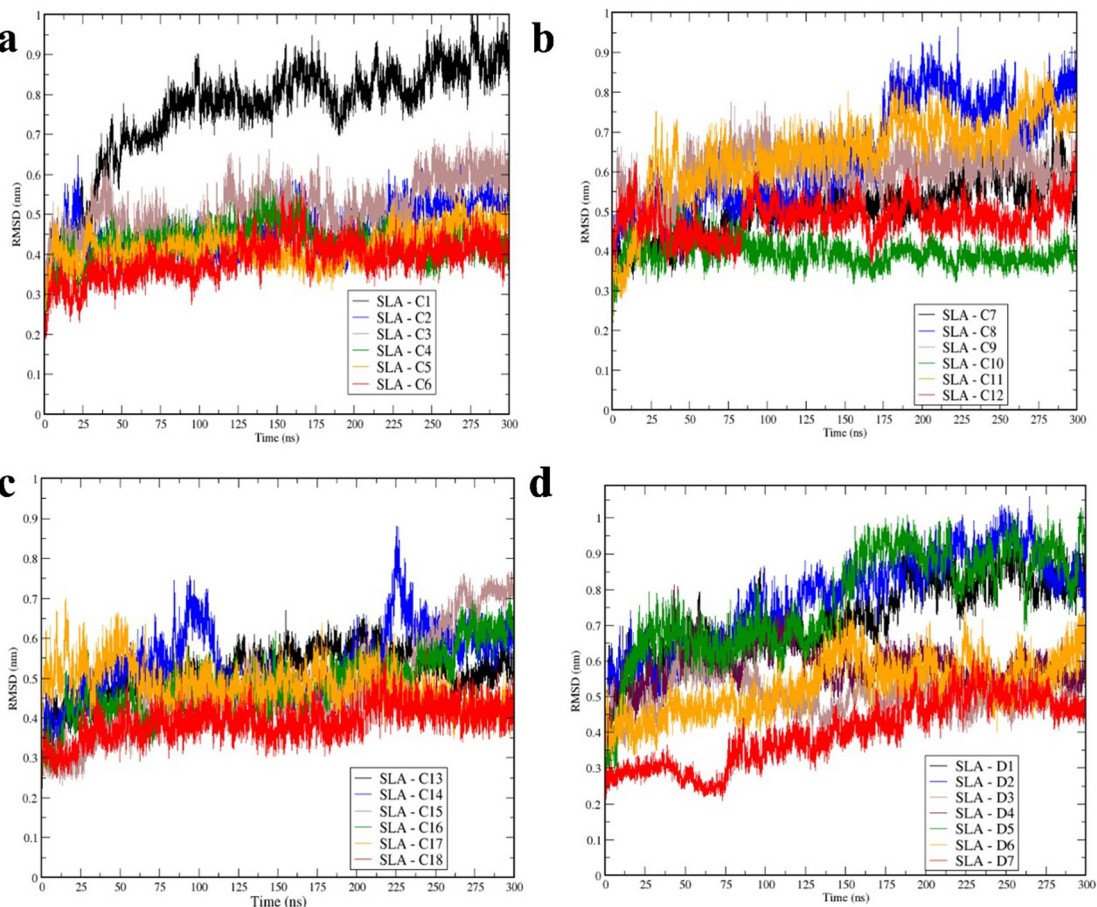

**Fig 3. Root mean square deviation (RMSD) from molecular dynamic simulation.** (**a**) RMSD for SLA-1 simulations C1—C6. (**b**) RMSD for simulations C7 – C12. (**c**) RMSD for simulations C13 – C18. (**d**) RMSD for SLA-2 simulations of D1 – D7. (**a—d**) Graphs are colored and sorted based on the number (C1 – C6, C7-C12 and C13-C18) from black, blue, brown, green, orange and red, respectively for (a-c) and (D1-D7) are represented by black, blue, brown, maroon, green, orange and red (d).

that our complexes are stable in most simulations. Selection of strong CTLEs only based on SASA or solely by considering the RMSD is possible but might not give the appropriate estimation of the binding. Therefore, we opted to consider free energy evaluation between the CTLEs and the 2 SLA complexes as the basis for our final CTLE selection for use in our vaccine (**Fig 5 and S4 Table in S1 File**).

Free energy of binding between SLAs and each CTLE was calculated by using *g_mmpbsa* tool as detailed in the Materials and Methods section [42]. The free binding energy for C2, C3, C8, C9, C11, C12, C13, C14, C15, D2, D4, D5, D6 and D7 show highly favorable binding between the CTLE and SLAs. Simulations with SLA-CTLE binding energy of > -100 kJ/mol were observed for D2, and D4 which were -137.8 kJ/mol and -104.2 kJ/mol, respectively (**Fig 5**). The strongest binding energies were calculated for D5, C9, D7, D6 and C12 at -305.6 kJ/mol, -219.5 kJ/mol, -214.8 kJ/mol, -185.1 kJ/mol and -139.5 kJ/mol, respectively, indicating strongly favorable interactions between these CTLEs and the SLAs. Other simulations, including C1, C4-C7, C10, C16-C18, D1 and D3 generated non-favorable binding with SLA-1 or SLA-2 and they were excluded as it was assumed they would not be effective CTLEs. By offering a detailed thermodynamic understanding of the forces driving epitope binding, free energy

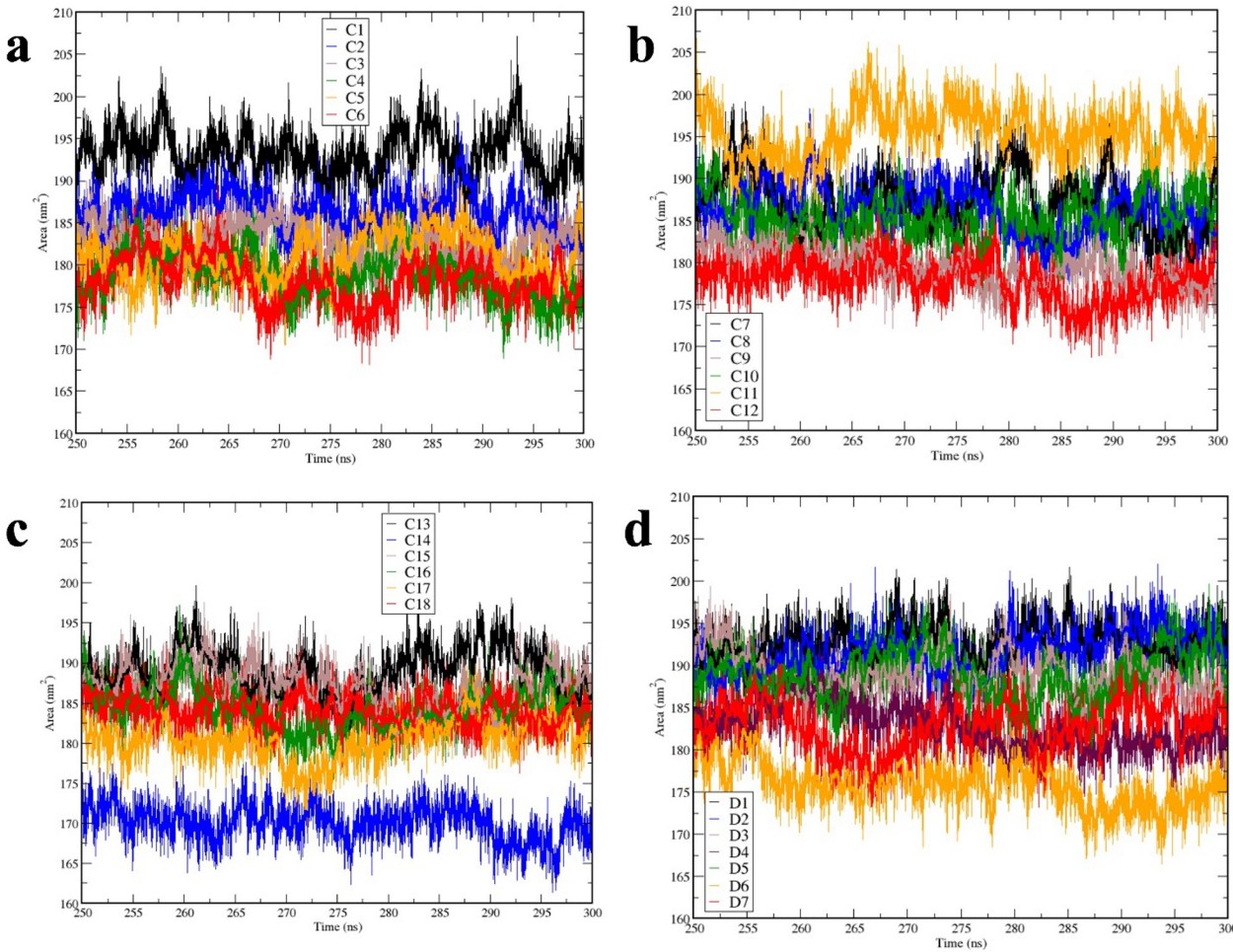

**Fig 4. Solvent Accessible Surface Area (SASA) calculations from 250 ns to 300 ns.** (**a**) SASA for SLA-1 simulations, C1—C6. (**b**) SASA for simulations C7 – C12. (**c**) SASA for simulations C13 – C18. (**d**) SASA for SLA-2 simulations of D1 – D7. (**a—d**) Graphs are colored and sorted based on the number (C1 – C6, C7-C12 and C13-C18) from black, blue, brown, green, orange and red, respectively for (**a-c**) and (D1-D7) are represented by black, blue, brown, maroon, green, orange and red (**d**).

evaluations enhance the precision of epitope predictions, improving the overall efficiency of vaccine design.

In our next analyses, we calculated the average contribution of residues in α1 and α2 regions of the SLA in binding energy to the top 14 CTLEs. The contributions (favorable or non-favorable) of all residues were assessed, then the sum of binding was calculated and the residues with favorable binding in selected simulations of C2, C3, C8, C9, C11, C12, C13, C14, C15 (**Fig 6A**) and simulation of D2, D4, D5, D6 and D7 are shown (**Fig 6B**). Note that we opted to only show residues with favorable participation (i.e., negative binding energy; **Fig 6**) for clarity. Several residues had negative binding energies in several simulations, but their sum was not negative, and they were therefore excluded from the graph. Several amino acids in the α1 region for SLA-1 and SLA-2 showed strong binding energies to CTLEs: Glu148 in C9 and C13 (with -39.6 kJ/mol, -18.7 kJ/mol), Asp151 in C9 and C13 (with -33.8 kJ/mol and -22.8 kJ/mol), Asp122 in C11 (with -19.7 kJ/mol), Trp147 in C14 (with -13.8), Glu58 in C3 (with

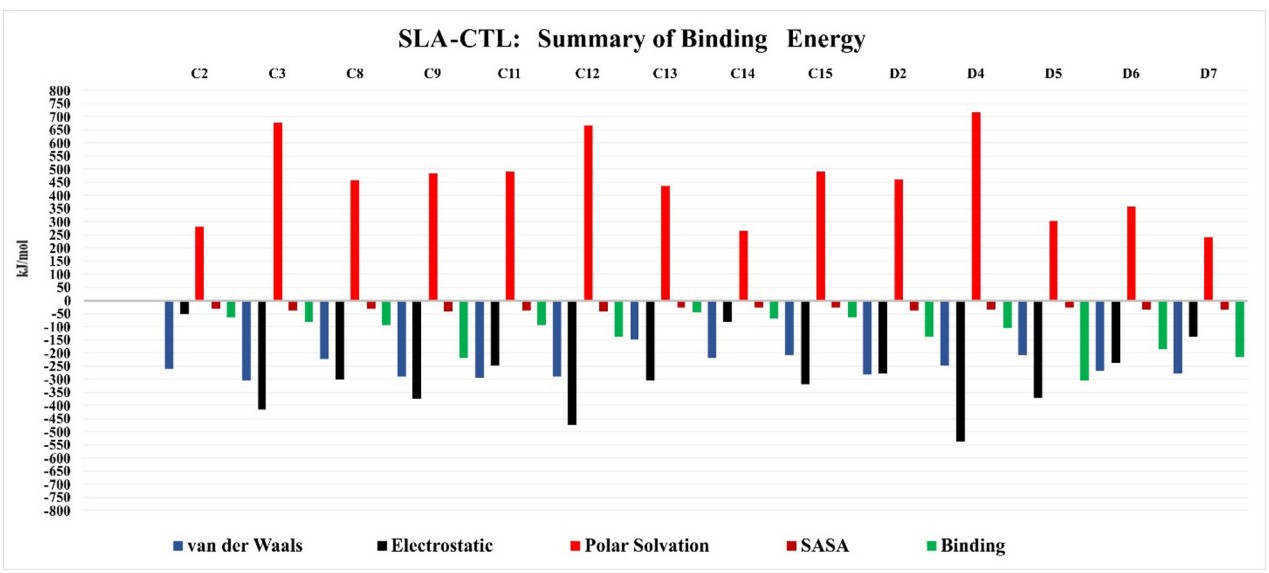

**Fig 5. The summary of binding energy between CTLE and SLA-1 and SLA-2.** The van der Waals, electrostatic energy, polar solvation and SAS energy readouts are colored in blue, black, red and dark red, respectively, each in kJ/mol. The free binding energy is shown in green. All evaluated energy data and the standard deviation for each value can be found in S3 Table in S1 File.

-13.5), Asp 122 in D5 and D6 (with -28.4 and -16.4). **Fig 6C** shows a representative crystalized 3QQ3 tertiary structure for an α1 region (magenta helix and strands) and an α2 regions (orange helix and strands) showing favorable contact with a CTLE with select amino acids (dark green). The lighter green helix and strand structure indicates the α3 region of the SLA which are not involved directly with CTLE peptide binding.

The dynamic nature of the CTLEs and the peptide binding grooves are thus captured using MD simulation and free energy calculations over time by monitoring of interactions between all molecules in the atomistic level. Computational simulation alongside the immunoinformatic methodologies can help find strong and potent epitopes that bind MHC/SLA proteins.

## B-cell epitope predictions

To identify the optimal linear B cell epitopes from these antigens to be used as part of our vaccine construct, the proteins were subjected to BepiPred- 2.0 [60], Bcepred (https://webs.iiitd. edu.in/raghava/bcepred/index.html) and ABCpred [61]. The generated B cell epitopes are shown in **S5-S8 Tables in S1 File**.

**Table 4** shows all high ranked B-cell epitopes for all four proteins. The 3D conformations of the antigens were modelled by Robetta [48, 49], and Protein Homology/analogY Recognition Engine V 2.0 (Phyre2) [62].

## Subunit construct development

We designed a protein comprised of the 5 tops ranked CTLEs (Table 3) and the top ranked B cell epitopes (Table 4) to be used as a multi-epitope vaccine construct. The gene was designed to be codon-optimized for expression in *E. coli*. The N-terminus (N-ter) of the resultant construct was comprised of 40 amino acids from Uniprot identifier Q1MRS4 as it shows strong binding affinity to TLR-5 interacting domain. It was followed by an EAAAK linker and C9, C11, C12, D5and D7 all separated by AAY linkers. Next, B1 epitope was linked to GGGGG linker, followed by B2-B3 epitopes which were separated by KK linkers. The linker regions

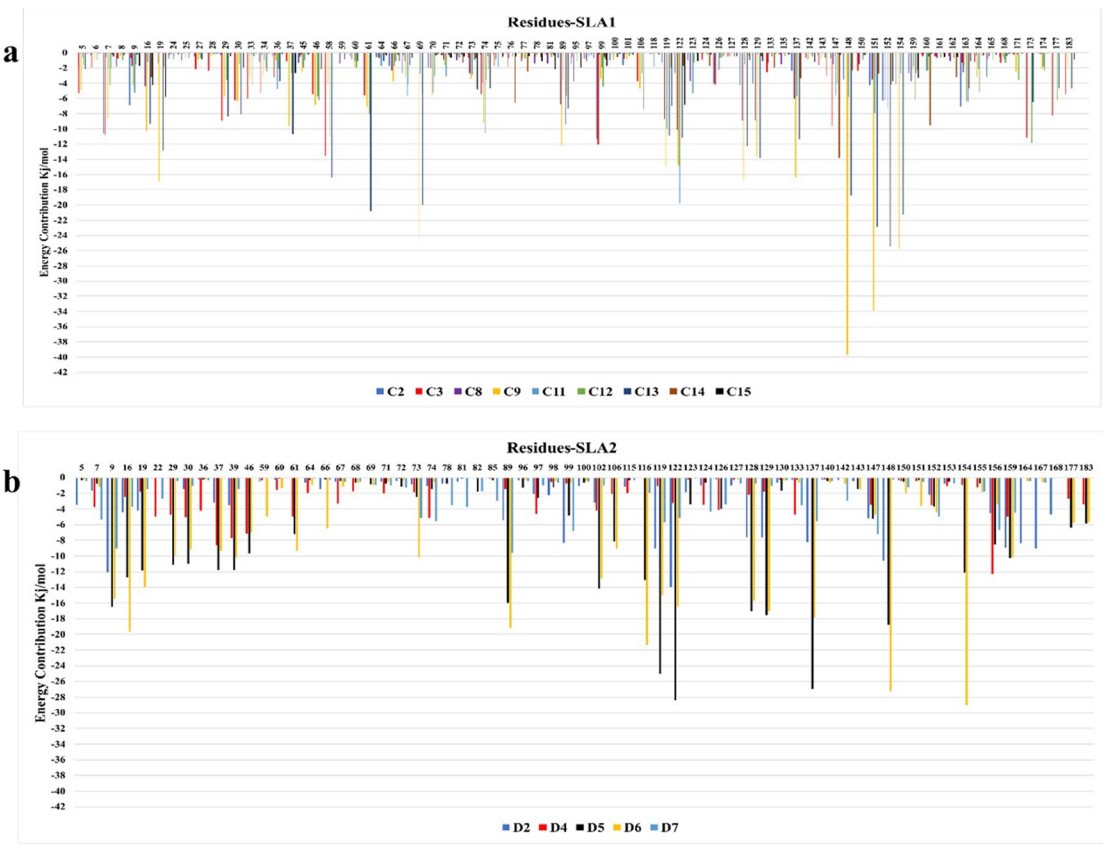

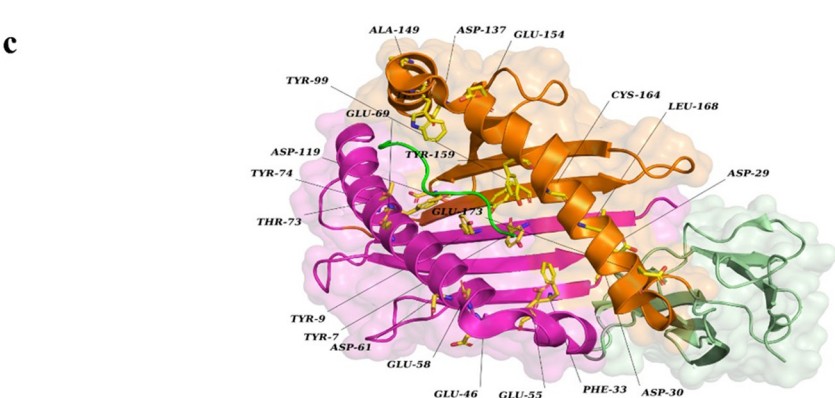

**Fig 6. The energy contribution between SLA residues and their docked epitopes are measured for only representative simulations.** The X axis shows the simulations in which their energy is calculated, and the position of the residues at SLA conformation while Y axis represents the value for energy contribution. (**a**) Binding energy between residues of SLA-1 and CTLEs at C2, C3, C8, C9, C11, C12, C13, C14 and C15 simulations. (**b**) Binding energy between residues of SLA-2 and CTLEs at D2, D4, D5, D6 and D7 simulations. All residues have both positive and negative contributions but only the one with negative energy contributions (i.e. favorable for binding) are depicted. (**c**) A representative tertiary structure for SLA-1 3QQ3 crystalized conformation with its CTLE. The green loop shows a CTLE. The magenta indicates the helix and beta-sheet for α1, and the orange sequence indicates helix and beta-sheet for α2 from SLA-I 3QQ3. The lighter green helix and strand structure indicates the α3 region of the SLA.

allow flexibility with no negative impact on conformations of the expressed protein. Indeed, it is crucial to highlight the importance of selecting the most appropriate combination of linkers

**Table 4. Linear B-cell epitopes.**

| Protein | Position | B-cell Epi | Methods |
|---------|----------|------------|---------|
| Q1MPW1 | 5 | IGIDLGTTNSCVYV | BepiPred-2.0 |
| Q1MPW1 | 395 | IETMGGVFTKLIDRNTTIPTR | BepiPred-2.0 |
| Q1MPW1 | 484 | DLGTGKEQSIQITASSGLS | BepiPred-2.0 |
| Q1MPW1 | 597 | YKQTQETSGASGDPTDTSA | BepiPred-2.0 |
| Q1MPW1 | 15 | CVYVMEGKEPKCITNP | ABCpred |
| Q1MPW1 | 171 | PTAASLAYGFDKKANE | ABCpred |
| Q1MPW1 | 310 | TIEPCSKALEDAGLQT | ABCpred |
| Q1MPW1 | 616 | SSSKSGDDVVDADFTE | ABCpred |
| Q1MPW1 | 28 | TNPNGGRTTPSVVAFTDK | BcePred |
| Q1MPW1 | 251 | MALQRLKDSAENAKKEL | BcePred |
| Q1MPW1 | 595 | QLYKQTQETSGASGDPTDTSASSSKSGD | BcePred |
| Q1MQM4 | 59 | ADSAVGPNPIASTHLTISTT | BepiPred-2.0 |
| Q1MQM4 | 163 | QQQPNEDQLVGGDININLEN | BepiPred-2.0 |
| Q1MQM4 | 222 | SNIATVMLGSHYDTTMAVGG | BepiPred-2.0 |
| Q1MQM4 | 788 | SGDFAAKSEVLNMKFKDKND | BepiPred-2.0 |
| Q1MQM4 | 112 | VENTNTQNSIIGGSMANA | ABCpred |
| Q1MQM4 | 526 | KGLIWSDIIFNPQDKT | ABCpred |
| Q1MQM4 | 828 | HLDIFGDLGNDKGIGGQV | ABCpred |
| Q1MQM4 | 2 | AYLSISKNQC | BcePred |
| Q1MQM4 | 682 | QFATNRTKTKC | BcePred |
| Q1MQM4 | 798 | LNMKFKDKNDT | BcePred |
| Q1MPM8 | 36 | SSPETGVPASMQWWKR | BepiPred-2.0 |
| Q1MPM8 | 103 | TPVWVDHKRVTDGQSPYS | BepiPred-2.0 |
| Q1MPM8 | 245 | PRLIMDTAIERGVSMKDLS | BepiPred-2.0 |
| Q1MPM8 | 315 | GIVSPHLSDLLKNP | BepiPred-2.0 |
| Q1MPM8 | 44 | ASMQWWKRFNDSTLDI | ABCpred |
| Q1MPM8 | 196 | TAQYQKGFINKLDLTRAK | ABCpred |
| Q1MPM8 | 7 | CIITCVIVSSCSFAPDYN | ABCpred |
| Q1MPM8 | 437 | ASAWSDRLSSIVQVCL | ABCpred |
| Q1MPM8 | 33 | VWVSSPE | BcePred |
| Q1MPM8 | 180 | IAERTVK | BcePred |
| Q1MPM8 | 353 | EAAQAKE | BcePred |
| Q1MPM8 | 438 | SAWSDRL | BcePred |
| Q1MRS4 | 78 | SLFTPEDKQKIFF | BepiPred-2.0 |
| Q1MRS4 | 91 | ITCPSCKKEVAYNVE | BepiPred-2.0 |
| Q1MRS4 | 47 | AFKIFTEGTSLEGAIL | ABCpred |
| Q1MRS4 | 29 | VKVCYGELTNIVPDSLQF | ABCpred |
| Q1MRS4 | 79 | LFTPEDKQKIFFITCP | ABCpred |
| Q1MRS4 | 13 | IIQEEMSKNGV | BcePred |
| Q1MRS4 | 78 | SLFTPEDK | BcePred |
| Q1MRS4 | 91 | ITCPSCK | BcePred |

The amino acids with scores above the threshold value (default value is 0.5 for BepiPred and above 0.8 for Bcepred and ABCpred) are shown.

or modifying the N-terminal protein to optimize experimental outcomes. We could also suggest using flagellin in the N-terminal site as it is a known TLR5 agonist as well [63]. The flexibility to customize these elements can significantly impact the success of trials. A graphical

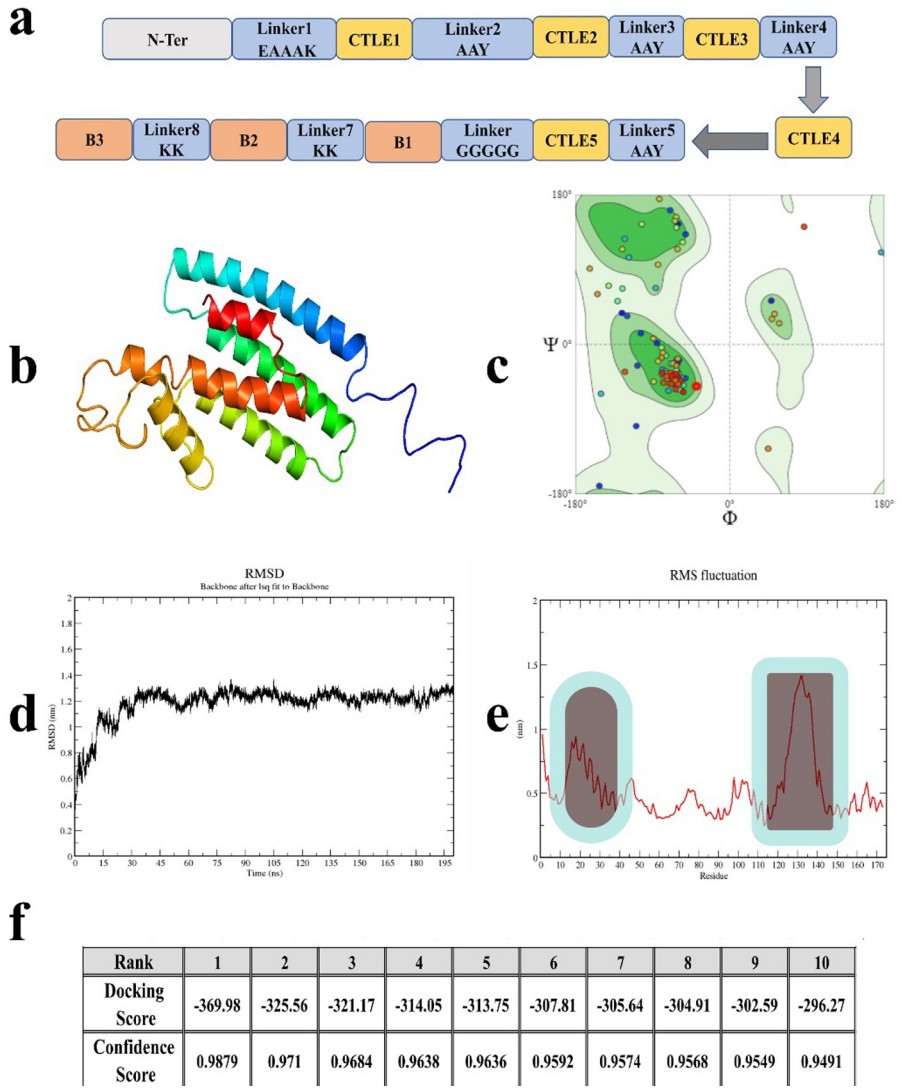

**Fig 7. Vaccine construct.** (**a**) The arrangement of B and CTL epitopes in the construct from N to C terminal. (**b**) The 3D conformation of the construct. (**c**) The general Ramachandran plot with more than 94% favorable. (**d-e**) RMSD and RMSF (Root mean square fluctuation) of the construct. (**f**) Docking results between swine Toll-like receptor 5 (T1UMR1_PIG) and the construct for top ten generated models.

representation of the protein construct is depicted in **Fig 7A**. The construct was modelled by Robetta (**Fig 7B**), and results showed that the secondary structure includes > 70% of helices, turns and the remining regions consist of loops that connect different helices to each other with acceptable Ramachandran scoring of > 94% favorable for general residues (**Fig 7C**). We used the MM/PBSA for free energy evaluations, which provide a reliable estimation of binding between studied sets of molecules [64–66]. We conducted the MD simulation for the recombinant vaccine antigen and the RMSD analyses showed a maximum of 1.2 nm, indicating some structural changes occurred over time but because the RMSD data stabilized at the end of the MD simulation, it suggests that the recombinant protein's conformation was relatively stable (**Fig 7D and 7E**). The analysis of structural motion of our vaccine construct indicates that residues 12–27 at the N-terminus show relatively moderate flexibility with high fluctuations around 1 nm, while residues 121–142 show higher mobility with fluctuations around 1.5 nm,

suggesting that these regions are more dynamic. These types of analyses were performed to indicate protein stability which impacts protein function. However, because this recombinant protein is an antigen, it does not have 'functional/active' regions. The relatively dynamic nature of the C terminal region may be advantageous to allow for B cell receptor binding to the B cell epitopes, but the CTLEs in the antigen will be processed by the proteasome in the antigen presenting cell and therefore whether these regions are linear or have a secondary structure is not critical.

Many studies that apply computational modelling and MD simulations in immunoinformatic analysis of antigens evaluate the interactions of a vaccine antigen with innate immune response receptors, such as Toll-like receptor 5 (TLR-5) through MD simulation. It is not clear why these analyses inform on the predicted efficacy of the vaccine as TLR-5 is a pattern recognition receptor that recognizes pathogen-associated molecular patterns such as flagellin. The vaccine antigen would, instead, be taken up by antigen-presenting cells, the cells will migrate to secondary lymphoid organs and present the epitopes from the vaccine antigen to T cells on the SLA complexes. Likewise, the vaccine antigen will be bound by B cell receptors on the B cells, internalized and then presented onto SLA complexes to become activated by Th2-effector T cells. However, we included this comparison with TLR5 (Uniprot: T1UMR1_PIG) using HDOCK tool to be consistent with other work in the literature. The HDOCK tool ranks the best poses and assigns them the energy score based on it scoring function. Here, the first pose is selected and all residues at the vicinity of 8Å relative to the TLR-5 were selected. The sum of first five docking poses was ∼ -1644.5 kJ/mol, which shows relatively strong binding between the peptide construct and swine TLR-5 (**Fig 7F**). The results indicate that our protein has relatively strong affinity to bind with TLR-5.

Finally, the physicochemical characteristics of the vaccine construct were calculated. Using Protein-Sol, we calculated the predicted solubility of the vaccine construct to be 0.61 when compared to the population average for the experimental dataset score of 0.45. These data suggests that it is soluble when over-expressed, the antigenicity was examined by http://scratch.proteomics.ics.uci.edu/ and data showed that predicted probability of antigenicity of 0.93. Its pI is predicted to be 6.0 and its MW is 18837.3 calculated by Expasy https://web.expasy.org/cgi-bin/compute_pi/pi_tool. Vaxijen v2.0 software predicted that the antigenicity of the vaccine construct was 0.79 which is significantly higher than its standard threshold score of 0.4. Our predicted construct shows strong antigenicity, high solubility, strong affinity towards TLR-5 and favorable Ramachandran plot as an indication of conformational integrity. These results suggest that the recombinant protein can be expressed well in *E. coli* expression systems.

Current vaccines against LI are either derived from killed bacteria (Porcilis Ileitis, Merck) or live attenuated bacteria (Enterisol, Boehringer Ingelheim) and there is no effective alternative subunit vaccine in the market. CTLE-based vaccines against LI infection are a rational strategy as the cell-mediated immune response has been shown to be important for protecting pigs against LI infection [67, 68]. Therefore, we see tremendous value in identifying T-cell epitopes that can strongly bind to SLA to activate CD8[+] cells combined with B cell epitopes to use as a vaccine antigen. A subunit vaccine based on multiple B and T cell epitopes allows for targeting of multiple antigens and it allows for differentiation from infected versus vaccinated animals, as antibodies or cell-mediated immunity against antigens not included in the vaccine can be used to establish infection.

Accurately predicting which bacterial antigens have can trigger a robust T cell response has traditionally relied on performing costly vaccine experiments with outer membrane proteins, flagellins, or other candidate antigens. Reverse vaccination combined with computational modelling has been employed to screening and in silico validation for developing an effective

vaccine construct [69, 70]. These approaches have unveiled critical immunodominant epitopes where prediction by cloning and expression might be difficult. By using immunoinformatic approaches, antigens and epitopes expected to perform poorly are removed from further experimentation if they contain any regions in their sequences that causes toxicity or induce severe allergic reactions after injection. Subunit vaccine development through reverse vaccination is scalable and relatively economical making this very appealing to pharmaceutical development. During the last decade reverse vaccination resulted in significant scientific discoveries and applicable vaccines for a limited number of diseases [71, 72]. Thus, in this work, we employed several well-tested immunoinformatic tools as well as computational modelling, molecular docking, MD simulations, free energy calculations as a solely *in silico* approach to rapidly screen optimal candidate T and B cell epitopes from the LI proteome and for the design of multiepitope vaccine construct. We view this work as a pilot for large scale MD simulation at the scale of $> 1000$ for LI protein in close future. New advances in computational fields have overcome many limitations that benefit vaccine development but also development of immune therapeutics for a diverse range of diseases including. The next steps will be to perform vaccine trials in pigs to validate antibody binding studies with the peptide antigen as well as assess how well the CTLEs promote a cell-mediated immune response.

Furthermore, while the current work targeted CTLEs to bind to only two SLAs, it is strongly recommended that researchers use at least 10% of available SLA in their pipeline to ensure that a herd member with distinct SLAs can also generate a robust T cell response to the predicted CTLEs. We stipulate that the significance of genotyping SLAs or performing RNA sequencing experiments in the national and international swine herds will be critical to find appropriate SLAs to target.

## Supporting information

**S1 Fig. Conformations of experimentally validated epitopes that are deposited in the Protein data Bank (PDB).** It shows that all epitope obtaining the loop shape while they in interatom with their binding site of the respective MHC receptors.
(TIF)

**S2 Fig. The conformations of the generated epitopes through this study are resembling the same conformations of the crystal structures.** We only show C1-C6 (A-F) and D1-D3 (G-I)., this is applying to all other conformations.
(TIF)

**S3 Fig.** The Ramachandran plot for our epitope-SLA for simulation C1-C6 (A-F) and D1- D3 (G-I).
(TIF)

**S1 File. Supplementary tables.** S1-S8 Tables are included herein.
(DOCX)

## Author Contributions

**Conceptualization:** Zahed Khatooni, Heather L. Wilson.

**Data curation:** Zahed Khatooni.

**Formal analysis:** Zahed Khatooni.

**Funding acquisition:** Heather L. Wilson.

**Investigation:** Zahed Khatooni.

**Methodology:** Zahed Khatooni.

**Project administration:** Heather L. Wilson.

**Resources:** Sanjeev K. Anand.

**Supervision:** Heather L. Wilson.

**Writing – original draft:** Zahed Khatooni.

**Writing – review & editing:** Gordon Broderick, Heather L. Wilson.

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
