## [Decision Letter · Decision Letter 0]

22 Aug 2024

PONE-D-24-30473Combined computational biochemistry and immunoinformatic approaches for development of subunit-based vaccine against the Lawsonia intracellularisPLOS ONE

Dear Dr. Wilson,

Thank you for submitting your manuscript to PLOS ONE. After careful consideration, we feel that it has merit but does not fully meet PLOS ONE’s publication criteria as it currently stands. Therefore, we invite you to submit a revised version of the manuscript that addresses the points raised during the review process. Please submit your revised manuscript by Oct 06 2024 11:59PM. If you will need more time than this to complete your revisions, please reply to this message or contact the journal office at plosone@plos.org. Please include the following items when submitting your revised manuscript:A rebuttal letter that responds to each point raised by the academic editor and reviewer(s). You should upload this letter as a separate file labeled 'Response to Reviewers'.A marked-up copy of your manuscript that highlights changes made to the original version. You should upload this as a separate file labeled 'Revised Manuscript with Track Changes'.An unmarked version of your revised paper without tracked changes. You should upload this as a separate file labeled 'Manuscript'.

We look forward to receiving your revised manuscript.

Kind regards,

Rajesh Kumar Pathak, Ph.D.

Academic Editor

PLOS ONE

Journal Requirements:

5. We note that you have indicated that there are restrictions to data sharing for this study. PLOS only allows data to be available upon request if there are legal or ethical restrictions on sharing data publicly. For more information on unacceptable data access restrictions, please see http://journals.plos.org/plosone/s/data-availability#loc-unacceptable-data-access-restrictions. 

Additional Editor Comments:

The manuscript is of significant interest and holds great potential. However, a thorough revision is recommended to address the reviewers' concerns. Specifically, the manuscript would benefit from clarifying the methodology and strengthening the rationale behind the key computational approaches used. These revisions will help ensure that the manuscript meets the high standards required for publication.

Reviewers' comments:

Reviewer's Responses to Questions

**Comments to the Author**

1. Is the manuscript technically sound, and do the data support the conclusions?

Reviewer #1: Yes

Reviewer #2: Partly

2. Has the statistical analysis been performed appropriately and rigorously? 

Reviewer #1: No

Reviewer #2: No

3. Have the authors made all data underlying the findings in their manuscript fully available?

Reviewer #1: Yes

Reviewer #2: Yes

4. Is the manuscript presented in an intelligible fashion and written in standard English?

Reviewer #1: Yes

Reviewer #2: Yes

5. Review Comments to the Author

Reviewer #1: The author should primarily define the bacteria (gram positive or negative) it help to the readers, researchers and annotators in broad spectrum

As authors states in abstract "free energy calculations in a novel approach at three stages to find strong T and B cell epitopes in LI proteome.", the method is already known and well cited so the sentence could be reframed.

Significance

Statement by Authors:

Importantly, this is the among few study that extensively employed computational biophysics

and combined it to reverse vaccination

for developing a subunit vaccine against one of the most dangerous bacteria in the meat industry.

There are many studies in other sectors, please find here few (Suggestions to modify the sentence)

https://pubmed.ncbi.nlm.nih.gov/37499394/

https://pubmed.ncbi.nlm.nih.gov/36311718/

Title: 2.4.1 CTLE Docking and Structural prediction by Homology Modelling - is not appropriate way

Kindly reframe for appropriate way

Structure prediction by homology modelling, structural assessment, binding site selection and molecular docking

Structural assessment is missing, an important part in structure validation kindly add (at least Ramachandran Plot)

Reviewer #2: In this manuscript titled ‘Combined computational biochemistry and immunoinformatic approaches for development of subunit-based vaccine against the Lawsonia intracellularis,

The authors report an innovative in-silico approach for designing a multi-based vaccine against the Lawsonia intracellularis. They concluded that the in-silico approach is suggested to have the potential to induce a robust immunological response against Lawsonia intracellularis in piglets. While the topic is important for public health and of general interest to the scientific community, the manuscript should be thoroughly revised before it can be considered for publication.

Comments and Clarification :

1. Why is the specifically GGGGG linker used to connect CTLE and B-cell epitopes?

2. What new insights does this study offer compared to previous research?

3. How do extended MD simulations and free energy evaluations improve epitope prediction accuracy over traditional methods?

4. Why are overexpressed and functionally unknown LI proteins prioritized for vaccine development?

5. What criteria were used to select 19 LI proteins from an initial pool of 256?

6. Why is allergenicity not considered in this study?

7. Which tool was used to validate the structure?

8. Why sum the binding energies of the top five docking models?

6. PLOS authors have the option to publish the peer review history of their article (what does this mean?). If published, this will include your full peer review and any attached files.

Reviewer #1: No

Reviewer #2: No

---

## [Author Response · Author response to Decision Letter 0]

16 Oct 2024

Response to Reviewer 1:

First, we would like to thank you for critically reviewing our work:

1. You indicate that there are many studies that that employed similar approaches as ours and you cite two examples:

https://www.sciencedirect.com/science/article/pii/S156757692300961X?via%3Dihub

In this work, the authors performed three MD simulations for the Spike protein from three different strains of the SARS-CoV-2 virus. We appreciate this work but would like to point out that these MD simulations are showing protein structure in isolation. Our work shows how several CTLEs interreact with the peptide binding groove from 2 SLAs for the purposes of validating binding affinity without negatively impacting peptide binding groove overall conformation. Our approach shows that 300 ns MDS is needed to ensure excellent binding affinity and therefore contributes to epitope prediction.

https://www.frontiersin.org/journals/immunology/articles/10.3389/fimmu.2022.1042997/full

In Figure 5 of the 2nd cited work, they used MD simulation of their vaccine construct, which is similar to what we applied in Figure 7. They showed that MDS can provide value in predicting protein conformations. Again, we used MDS to validate predicting binding affinity between the epitope to the peptide binding groove of relevant SLA complexes, not simply assessing the overall predicted structure of a vaccine antigen. We then show that extended binding of the epitope to the SLA did not negatively impact the shape of the peptide binding groove, which could be missed if the MDS was only performed for picoseconds. 

We removed the ‘Significance section’ and instead inserted these sentences within the abstract:

“Collectively, these molecular modeling and immunoinformatic analyses led to the creation of a multi-epitope protein comprised of the top-ranked T and B cell epitopes against LI. We believe this work presents a useful in silico protocol for the discovery of candidate antigen in many viral and bacterial pathogens.”

2. Thank you, we have taken you suggesting and changed the subsection title to: 

Line 135: Structure prediction by homology modelling, structural assessment, binding site selection and molecular docking

3. We performed Structural Assessment of the vaccine construct, including a Ramachandran plot, in Fig 7 and we have now included Ramachandran plots on the top 25 CTLEs (See Supplementary Figure 1).

See Line 137 and Lines 318-323.

In Supplementary Figure 1 (SFig 1A), we show the conformations of crystalized epitopes in the Protein DataBank. Note that they all have a loop structure. In SFig 1B and C, we show the conformations of representative CTLEs and the corresponding Ramachandran plots. These structural assessments ensure that the selected CTLEs also in a loop conformation and that their generated structures were relaxed, and their residues are positioned in terms of allowed ϕ and ψ angles.

Response to Reviewer 2:

Thank you for critically reviewing our work:

1. Why is the specifically GGGGG linker used to connect CTLE and B-cell epitopes?

The GGGGG linker is extremely flexible, and we reasoned that if a flexible linker was used to connect the B and T cell epitopes, then the linker would not likely contribute to conformational interference. Secondly, this long flexible linker would also provide more room/less conformational restraint between the B cell receptor (antibody) binding sites and the rest of the antigen.

These studies show these GGGGG linker is flexible:

https://www.ncbi.nlm.nih.gov/pmc/articles/PMC7370533/

https://link.springer.com/article/10.1007/s12013-020-00912-7

2. What new insights does this study offer compared to previous research?

Thank you for the opportunity to better explain the novelty of our approach. 

To the best of our best knowledge, most studies only use the immunoinformatic methodologies to predict strong B and T cell epitopes from approx. 50 antigens - we use the entire LI proteome and narrow down to the top 286 antigens. Others tend to use NETMHCpan to predict approximately 50 T cell epitopes, and then generate a vaccine comprised of several epitopes separated by linkers. They may perform homology modeling for the vaccine construct to visualize how it will form in solution. Some studies use molecular docking to predict the binding between epitopes/antigens to TLRs, which we do not feel is relevant as the antigens should be presented onto MHC complexes, not act as innate immune (TLR) stimulators. They may then model their vaccine constructs using MD simulation for a picosecond simulation to show the structural dynamics of their vaccine constructs. 

No other study has been published that performs MD simulation for SLA and their bounded epitope coupled with free energy evaluation. We used extensive high-capacity computing hours to obtain free energy calculations using RMSD, SASA to predict these epitopes that bounded strongly to the grooves.

We rewrote the introduction to make it much clearer how we are using already known bioinformatic tools (Stages 1 and 2) to identify antigens, then predict the best T cell epitopes using NETMHCPan 41. In Stage 3, we perform homology modeling to predict the shape, structure and charge of the CTLEs. Next, we perform docking studies with the top ranked CTLEs with 2 SLA complexes to calculate the free binding energy as a mean to quantify binding affinity. Stage 3 also entailed utilizing molecular dynamics (MD) simulations to fully investigate the atomic-level dynamics of proteins under the natural thermal fluctuation of water and thus potentially provide deep insight into the CTLE-SLA interaction (Fig 1). 

We also restructured the manuscript to have a combined Results and Discussion section. During this revision stage, we included an interpretation of the results and highlighted its importance to the field.

• Lines 229-232: Despite this high degree of similarity, our homology modeling of SLAs has previously shown that while the conformation of the overall SLA and/or the peptide binding groove may appear subtle, the changes to the electrostatic contact mapping may be dramatically different with just a few amino acid differences (1). Therefore, homology modeling followed by molecular docking and MD simulation are critical to predict CTLE-SLA peptide binding groove affinities. 

• Lines 272-276: These results clarify how several factors can contribute to the predicted antigenicity of proteins within a bacterial proteome. At this time, there is no clear formula for predicting which antigens would contribute to an effective vaccine, but we used these online tools to give us best-guesses for promising bacterial antigens who’s predicted epitopes can be interrogated and characterized below. 

• Lines 308-313: These online resources describe predicted characteristics of the epitopes and tools such as NetMHCPan4.1 has its own mathematical formulas to predict binding affinity to only a limited number of SLAs and it can inform epitope selection. But again, there is no clear formula for predicting which epitopes will bind with strong affinity to MHCs, which is a critical component of predicting the strength of the resultant immune response. To predict binding affinity between the CTLE and the peptide binding groove, we needed to perform molecular docking and MD simulations.

• Lines 330-335: Thus, we used molecular docking to computationally achieve an optimized conformation and relative orientation between the CTLE and the SLA peptide binding grooves such that the free energy of the overall system was minimized. We have some confidence that the interaction has the potential to be stable and may, therefore trigger a strong T cell response (2).The top ranked CTLE-SLA partners determined to have the lowest free energy were used as input for subsequent MD simulations (3). 

• Lines 342-345: Our next steps required that we track the atomic interaction between the amino acids that comprise the CTLE and the peptide binding grooves over time. MD simulations refine initial docking models, ensuring that the epitopes adopt more biologically accurate conformations while revealing which interactions remain stable across the simulation period.

• Lines 347-349: To the best of our knowledge, this study is among the few works that applied large-scale MD simulations and free energy evaluation on SLA – CTLE complexes to identify strong CTLEs. We performed 300 ns analyses on the top 25 ranked epitopes predicted to interact with the binding-site of the indicated SLA.

• Line 401-403: By offering a detailed thermodynamic understanding of the forces driving epitope binding, free energy evaluations enhance the precision of epitope predictions, improving the overall efficiency of vaccine design.

3. How do extended MD simulations and free energy evaluations improve epitope prediction

Extended Molecular Dynamics (MD) simulations, coupled with the basic MD based analyses such as RMSD, RMSf and most importantly the free energy evaluations, can provide critical insights into the prediction of T cell epitopes by offering a detailed atomistic level information of their stability and binding behavior. We know that one of the most important and very critical steps for designing a very strong T cell epitope is the effort and tool that accurately predict the strength of binding of the epitopes into their binding site of targeted MHC. MD simulations allow us to track how epitopes interact with major histocompatibility complexes (MHC) over time, and to track which amino acids of the MHC are mainly involved in those interactions. It also identifies stable conformations by studying the dynamic movements. MD simulations refine initial docking models, ensuring that the epitopes adopt more biologically accurate conformations while revealing which interactions remain stable across the simulation period. 

Free energy calculations, including methods like MM/PBSA, helps us to rank epitopes based on their binding potential, highlighting strong candidates for inducing effective T-cell responses. By offering a detailed thermodynamic understanding of the forces driving epitope binding, free energy evaluations aid in the selection of epitopes that not only bind robustly but also maintain stability and functionality within the immune system. Combining extended MD simulations with free energy analyses enhances the precision of epitope predictions, improving the overall efficiency of vaccine design.

4. Why are overexpressed and functionally unknown LI proteins prioritized for vaccine development? 

Thanks for your question.

We obtained lists of proteins that are over-represented in pathogenic stains of LI relative to lab-attenuated, non-pathogenic LI. We reasoned that these proteins were likely virulence factors that contribute to the pathogenic success in triggering a disease state and therefore they may be excellent vaccine antigen targets.

Why unknown proteins?

As well as the proteins that are overrepresented in the pathogenic strain, we opted to include proteins with unknown function in our antigen repertoire. The reason was that these proteins will not be part of any patent protection which would make vaccine development possible. Further, we wanted to show that our approach does not need the crystal structure of the proteins to be know and that epitopes can be identified without knowing anything about structure or function of the protein.

5. What criteria were used to select 19 LI proteins from an initial pool of 256? 

The reference proteome for the PHE/MN1-00 strain (uniprot.org/proteomes/UP000002430) is predicted to have ~ 1342 proteins.

We manually parsed the proteins to select those that are highly expressed in pathogenic LI strains based on research by Vanucci et al (4) or were proteins with unknown function (256 proteins listed in Supplementary (S) 1 Table).

The 256 selected LI proteins were then submitted to the VirulentPred to calculate the likelihood of the protein of being antigen based on Residue composition, Higher order Dipeptide Composition Based, PSI-BLAST-PSSM, PSI-BLAST created PSSM Profiles and Cascade of SVMs and PSI-BLAST (All based on VirulentPred).

They were then subjected to immunoinformatic assessments using a series of artificial intelligence and machine learning tools to determine expected virulence, solubility, antigenicity, and potential transmembrane regions (Table 1). 

1. One of these criteria was solubility, those protein that are more soluble have better chance to be exposed to immune system and this is an accepted approach in many studies to consider this factor for selecting the right antigen candidates. 

2. Second factor was antigenicity which was evaluated based on two different platform one was Vaxijen and another one based on the ANTIGENpro. These tools are selected when we have no solid knowledge of the protein function, and they can offer a prediction of antigenicity.

3. The third factor evaluated was looking if these proteins have a predicted transmembrane helices. If so, these antigens were removed as the amino acids would not be accessible for the B cell receptor nor will they be readily processed by the proteasome and loaded onto MHCs for presentation to T cell receptors.

6. Why is allergenicity not considered in this study?

Thank you for bringing this factor to our attention. We performed allergenicity studies using AllerCatPro.2 on the top CTLE epitopes. They did not show significant allergenicity.

Added to Line 305 and Line 132.

7. Which tool was used to validate the structure?

Thank you for your question:

First, generating the right conformation for the epitopes are one of the most important tasks to start this project. The first challenge is to identify confirmed conformations by assessing the crystalized (and therefore stable) SLA complexes with co-crystalized epitopes. We compare the conformations of these crystalized epitopes with predicted conformations of epitopes of interest and determined that they have a similar loop structure. Ramachandran plots also confirm that the epitopes are likely to be expressed (Supplementary Fig 1; Fig: 7).

Although we know that for each 3.6 amino acid in a peptide, there can be (possibility depends on forces) one turn/helix, we are confident that the peptides are not forming helices and are forming loops.

 Therefore, we have used several tools to develop the conformation of our epitope because the right conformation at first place was the main criteria for us. We found if we use CASB-Dock, it generated the right conformation for us which are loop, for more validation we have used the Ramachandran plot, for all complexes more than 95% of all residues are found in the allowed regions (Fig: C7b). This makes sure that we have started our simulation with the reliable conformation. 

8. Why sum the binding energies of the top five docking models?

While many studies only use one pose – the one with highest binding energy – we assessed differences between the 3D predicted confirmations and the known crystal conformations, and while they are loops, there are slight differences in structure. Therefore, several poses may be necessary to get a full idea of the CTLE conformation. For this reason, we opted to take the top 5 docking poses for each molecular docking. This approach improves reliability by accounting for variability in predicted poses, including alternative binding modes, and providing a more robust assessment of binding affinity.

---

## [Decision Letter · Decision Letter 1]

7 Nov 2024

Combined immunoinformatic approaches with computational biochemistry for development of subunit-based vaccine against Lawsonia intracellularis

PONE-D-24-30473R1

Dear Dr. Wilson,

We’re pleased to inform you that your manuscript has been judged scientifically suitable for publication and will be formally accepted for publication once it meets all outstanding technical requirements.

Kind regards,

Rajesh Kumar Pathak, Ph.D.

Academic Editor

PLOS ONE

Additional Editor Comments (optional):

The manuscript can be accepted now.

Reviewers' comments:

Reviewer's Responses to Questions

**Comments to the Author**

1. If the authors have adequately addressed your comments raised in a previous round of review and you feel that this manuscript is now acceptable for publication, you may indicate that here to bypass the “Comments to the Author” section, enter your conflict of interest statement in the “Confidential to Editor” section, and submit your "Accept" recommendation.

Reviewer #1: All comments have been addressed

Reviewer #2: All comments have been addressed

2. Is the manuscript technically sound, and do the data support the conclusions?

Reviewer #1: Yes

Reviewer #2: Yes

3. Has the statistical analysis been performed appropriately and rigorously? 

Reviewer #1: Yes

Reviewer #2: N/A

4. Have the authors made all data underlying the findings in their manuscript fully available?

Reviewer #1: Yes

Reviewer #2: Yes

5. Is the manuscript presented in an intelligible fashion and written in standard English?

Reviewer #1: Yes

Reviewer #2: Yes

6. Review Comments to the Author

Reviewer #1: The present work is scientifically appropriate, and will be support in vaccine development.

The authors have amended all the raised queries and appropriately answered.

The current manuscript can be further accept for publication.

Reviewer #2: (No Response)

7. PLOS authors have the option to publish the peer review history of their article (what does this mean?). If published, this will include your full peer review and any attached files.

Reviewer #1: No

Reviewer #2: No

---

## [Editor Report · Acceptance letter]

21 Nov 2024

PONE-D-24-30473R1 

PLOS ONE

Dear Dr. Wilson, 

I'm pleased to inform you that your manuscript has been deemed suitable for publication in PLOS ONE. Congratulations! Your manuscript is now being handed over to our production team.

Kind regards, 

on behalf of

Dr. Rajesh Kumar Pathak 

Academic Editor

PLOS ONE